# Sustainable Composites: A Review with Critical Questions to Guide Future Initiatives

**Martin A. Hubbe** 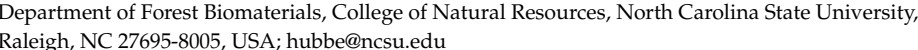

Department of Forest Biomaterials, College of Natural Resources, North Carolina State University, Raleigh, NC 27695-8005, USA; hubbe@ncsu.edu

**Abstract:** Composites, which have become very common in mass-produced items, have the potential to outperform similar materials made from any one of their individual components. This tutorial review article considers published studies that shine a light on what is required for such structures to earn the name "sustainable". The focus is on a series of questions that deal with such issues as the carbon footprint, other life-cycle impacts, durability, recyclability without major loss of value, reusability of major parts, and the practical likelihood of various end-of-life options. To achieve the needed broader impacts of limited research dollars, it is important that researchers choose their research topics carefully. Among a great many possible options for preparing truly eco-friendly composite materials, it will be important to focus attention on the much smaller subset of technologies that have a high probability of commercial success and large-scale implementation.

**Keywords:** sustainability; biocomposites; compostability; valorization; cascading

## 1. Introduction

A composite can be defined as a material composed of at least two major components that contribute in complementary ways to achieving desired performance attributes [1–3]. In many cases, one of the components has the role of a matrix, meaning that it forms an essentially continuous phase within the produced material. The other phase often can be called a reinforcing phase, and it may consist of discontinuous particles or fibers that are dispersed within the matrix phase.

The word "sustainable", as used in the title of this article, implies that a material such as a composite can be manufactured, used, and then either returned to the natural biological cycle or somehow recovered for a subsequent valuable purpose while minimizing any adverse impacts on the environment [4,5]. According to the United Nations statement of 1987, sustainability can be defined as "meeting the needs of the present without compromising the ability of future generations to meet their own needs" [6]. The most accepted approach with which to determine whether or not a certain manufactured product or process deserves to be called "sustainable" is to perform a life-cycle assessment (LCA), which attempts to quantify various types of impacts on the environment and other societal aspects [7,8]. When carrying out such an analysis, a specific proposed technological approach typically is compared against a chosen default approach. For instance, the currently most dominant production, usage, and recycling or disposal procedures may be assumed as the base case, against which a proposed technology is to be judged [9].

The goal of this tutorial review is to consider some critical questions, which may then serve researchers as guideposts and suggestions with respect to future research proposals and strategies. Though the questions and issues to be considered are not unique, they are often given too little emphasis. When basing one's decisions just on narrow criteria, there can be a danger of overlooking unintended consequences. For example, only a minority of the many possible new technologies that are proposed in LCA studies will have a high chance of being implemented at a large scale. The heterogeneous nature of

composite materials serves as a deterrent to many strategies of separation and recycling to be discussed in this review article [10,11]. The approach taken in this work will be to emphasize manufacturing options that have a high probability of large-scale usage in the coming years. The focus of this article is provided by a series of questions, with the goal of drawing direct attention to critical issues that call for rapid resolution. These questions deal with the following issues: the carbon footprint (effect on global warming potential); other life-cycle impacts; effects related to durability; circular economy issues; cascade principles; and the practical likelihood that a proposed technology will be implemented soon and at a large scale.

## 2. Does the Technology Have a Low Carbon Footprint?

### 2.1. The Global Warming Potential

The global warming potential (GWP) is a primary focus of many research articles that have addressed sustainability issues regarding composite materials. The term "carbon footprint" is often used to refer to the same concerns, since the net emission of carbon dioxide to the atmosphere is known to contribute to increases in average temperatures at ground level on the planet. In addition, the GWP contributions of other gases, such as methane and nitrogen oxides, are commonly expressed as $CO_2$ equivalents [12]. According to Mohsin et al. [13], the efficiencies of various processes generally have a dominant effect on net GWP averages when comparing different countries. Because GWP has profound effects throughout the whole world, it makes sense for researchers to place it high on their priority list when making decisions about what research projects to propose and carry out.

### 2.2. Reduction in Product Weight

As illustrated in Figure 1, in principle, there are two ways in which the incorporation of biobased materials, e.g., wood, into a composite structure might be expected to influence, and hopefully decrease, the value of GWP. First, the composite structure might increase the strength properties enough that less material would be required to meet the end-use requirements [14]. Second, the utilization of biobased content has the potential to decrease the GWP per unit mass of material in the mixture (see Section 2.3).

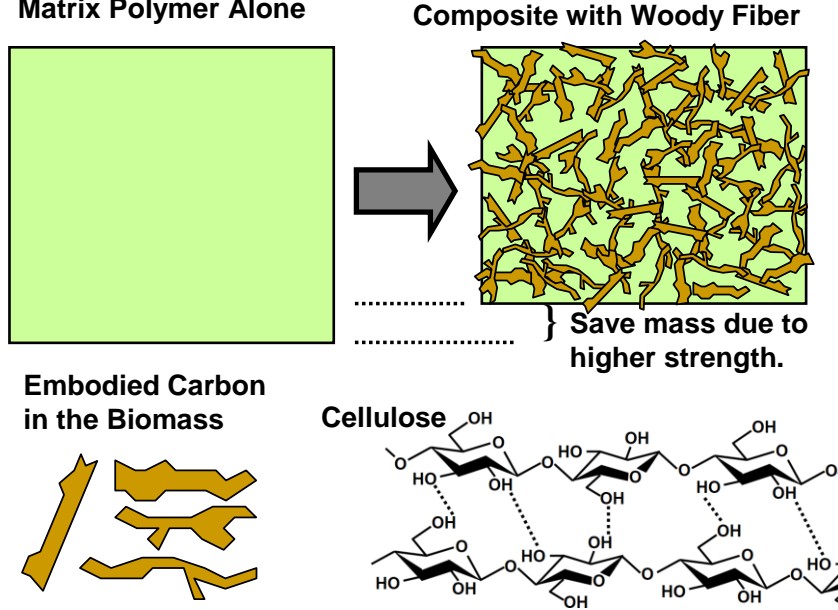

**Figure 1.** Concept of two ways that plant-based reinforcing fibers might contribute to environmental benefits: By achieving strength specification at lower product mass, thereby using less materials; or by containing carbon derived from photosynthesis (for example, carbon that is within cellulose).

Some ways of preparing fiber-reinforced composites have a lot of potential for achieving higher strength in comparison to the neat matrix polymer. This is especially true for certain glass fiber–epoxy and carbon fiber–epoxy systems, the strength of which greatly exceeds the strength of neat cured epoxy resin [15]. The cited work shows examples in which certain glass–epoxy and carbon fiber–epoxy composites achieved impact strengths in the range of 100 to 300 kJ/m², whereas glass–polypropylene composites achieved about 100 kJ/m². By comparison, typical composites with plant fibers and plastic matrices were in the range of 10 to 60 kJ/m².

The composites used in wind turbines and airplane parts achieve especially high increases in strength properties as a result of adding fibers—usually carbon fibers—to the continuous phase. Likewise, Ramesh et al. [16] highlighted certain plant-fiber-reinforced composite systems, involving surface treatments of the fibers, such that the flexural modulus exceeded that of a neat polyester resin. In such cases, less material might be needed, and some of the transportation expenses, including the use of energy in transportation, could be cut. It is worth noting, however, that increases in elastic modulus are typically easier to achieve in comparison with increases in tensile strength, by means of adding reinforcing particles to a plastic matrix [16,17]. In addition, any lack of compatibility at the fiber-to-matrix interface can serve as a starting point for brittle failure.

In systems where the reinforcing particles are mixed with the matrix, it is often found, that the addition of levels above a certain threshold, often in the range of 2% to 5%, leads to excessive clustering of the particles [17]. As illustrated in Figure 2, the combination of (a) mixing forces, (b) elongated particles, and (c) a high enough concentration of reinforcing particles is expected to lead to clustering. Examples of such reported findings or clustering, as well as the resulting decreases in strength when a certain threshold of concentration is exceeded, are given in the cited review article. The clustered fibers not only serve as a point of non-uniformity within the composite structure, but there is also often insufficient adhesive or binder between the clustered fibers. It also has been found that the most favorable strength improvements are achieved with fiber-like particles, but the relative improvement is typically no higher than a factor of two [17].

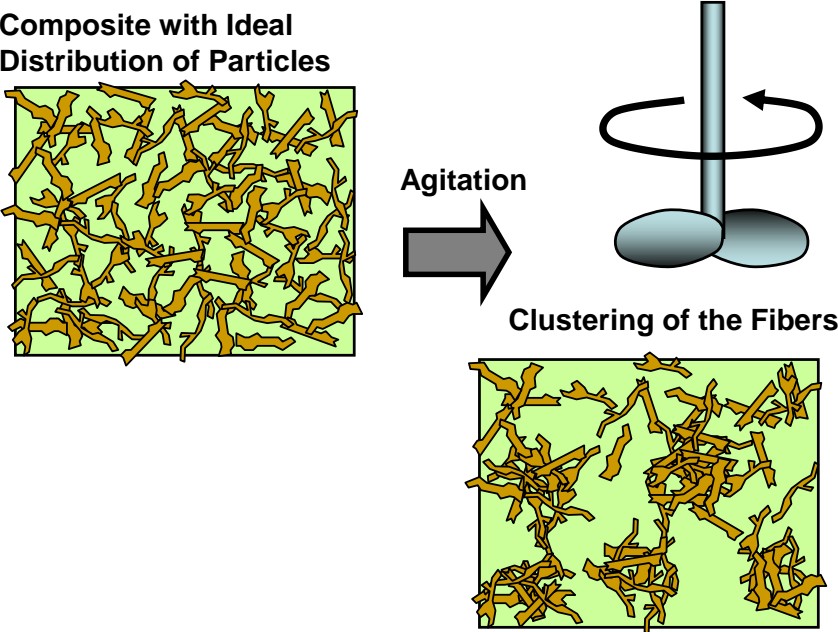

**Figure 2.** Illustration of a commonly observed clustering of reinforcing particles when stirring a mixture in melted polymer, having a high enough solids content that particle collisions are sufficiently frequent.

*2.3. Biobased Content*

Each component of a compost can be considered separately in an effort to quantify the overall GWP impacts associated with the preparation of a composite. Biobased components start out with a decided advantage, as they are formed during the process of photosynthesis. Since carbon dioxide is taken from the air during photosynthesis, the wood and other substances within a tree can be regarded as carbon-neutral. Thus, the mere inclusion of biobased components in a composite material is often taken as a favorable indication [18–20]. A more careful analysis, using an LCA approach, has to take into consideration such things as the usage of fossil fuels, either directly or in the form of electricity, during the planting and fertilizing of the plants, as well as the processing of the biomaterial and the preparation and initial transportation of the composite. It is often found that the inclusion of biomaterials, especially cellulose, will decrease the GWP of the final composite [20]. On the other hand, Correa et al. [21] found that the addition of kenaf fibers to a polypropylene matrix did not decrease the over carbon footprint when the energy required for processing the fibers was considered. Likewise, Lunetto et al. [22] found that glass-reinforced composites outperformed cellulose-fiber-reinforced composites with respect to energy requirements. Thus, it is important not to overlook the energy requirements associated with the growing, processing, and transportation of plant fiber material.

Options for the incorporation of plant materials as reinforcements in polymer matrices are numerous and diverse [2,3]. For example, plant fibers can be from seeds (e.g., cotton), or bast fibers (e.g., hemp), grasses, wood, etc., each being highly available and relatively low in cost [4]. The cited article lists hemp as one of the strongest fibers, making it a candidate for applications requiring strength. Khalid et al. noted that although it was possible to prepare "natural fiber reinforced polymer composites" with hemp fibers and a polymer matrix such as polyesters, their strength properties decrease with increasing immersion time in water [19]. The cited article describes many specific applications of sustainable composites, including commercial production.

LCA studies often reveal two ways in which the incorporation of biomaterials into composites tends to decrease the net release of greenhouse gases. In addition to incurring much less emission of such gases during manufacture, the composite product will embody a certain amount of biogenic carbon. Thus, as long as the composite material is in existence (not burned), that amount of carbon will not go back into the atmosphere.

## 3. Are Other Life-Cycle Impacts Low for Manufacture and Usage?

*3.1. Other Environmental Impacts*

A classical life-cycle assessment will consider not only the carbon footprint and GWT but also some other key issues affecting the environment. These will include the depletion of natural resources, the eutrophication of natural waters, ozone layer depletion, the acidification of the air and water, photochemical oxidation, and various toxicity effects [20]. These goals are illustrated in Figure 3. Some of these effects can lead to locally severe consequences, unlike greenhouse gases, which quickly become dispersed such that they can affect climate conditions throughout the world. Seile et al. [20] focused their LCA work on biobased composites, considering the impact of adding increasing amounts of hemp or flax fibers to a thermopolymer matrix. Strikingly, though the addition of the fibers to the composites yielded large reductions in $CO_2$ emissions, almost all of the other classes of environmental impact were negatively affected. Part of the explanation may lie in the requirements of fertilizer, as well as fuel for tractors, etc., during the growing and harvesting of cellulosic material. Note that a different conclusion might be reached if the fibers had come from corn stalks, i.e., the left-over parts of a food crop, for which the environmental impacts could be assigned, at least in part, to the food product [23,24]. In addition, the strengths of fibers from hemp or flax are known to be much higher than fibers from most agricultural residues [25,26].

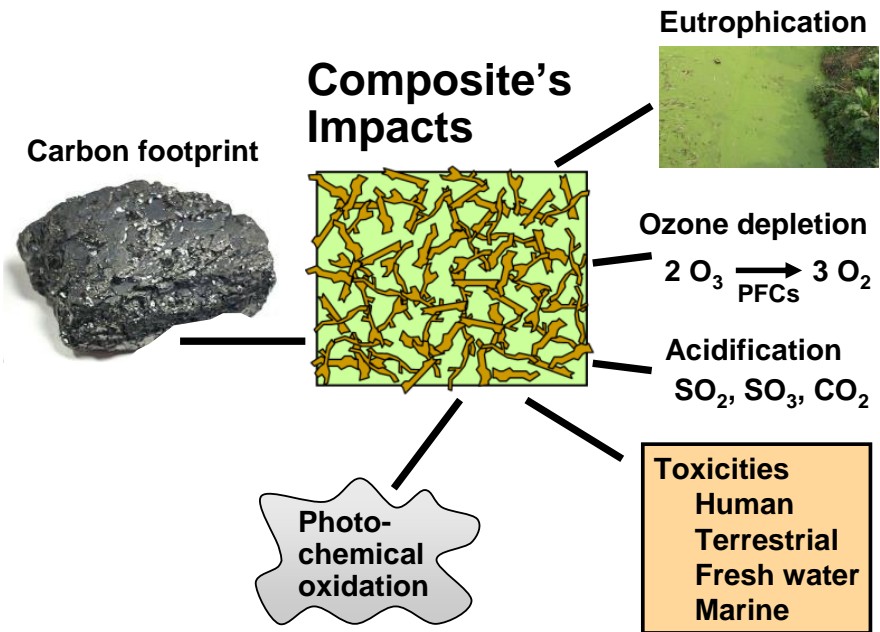

**Figure 3.** Multiple classes of environmental impacts, in addition to the carbon footprint (or global warming potential).

The relative importance attributed to the different categories in an LCA analysis is not fixed; rather, the weighting needs to be specified in each LCA report [27,28]. Because of the diversity of the categories, this is not a trivial task, and the selection can be highly subjective. Though it is possible to employ fuzzy analytic methods to assign the weights in an LCA analysis [28], the results of such approaches are rarely reported. Some items, such as ozone depletion, resemble the carbon footprint (and GWP impact), with respect to affecting the whole world. Others, such as eutrophication, might mainly affect a specific body of fresh water, in a given case. Policymakers often prefer a system in which a single overall number can be calculated, making it possible to compare the total global impacts of competing options. However, researchers need to bear in mind that any weighting systems and any summations involve arbitrary choices, not science, and that the goal is to keep each of the different impacts low enough to avoid significant environmental harm.

### 3.2. Economic Impact

In their review of composites that involve natural fibers, Asyraf et al. [5] defined three pillars of sustainability, namely ecology, economy, and social. Economic sustainability asks whether a certain technology can be profitable. Logic would suggest that technology options that are not affordable ought to be given an unfavorable overall rating, since they may be hardly worth paying attention to. The concept is illustrated in Figure 4. As shown, a load of financial costs is depicted as providing a barrier to the separation of a composite material back into its components. For example, the costs associated with such a separation might not be justified by the value of the recovered plastic (now partly contaminated and degraded) and the fibers (also partly contaminated and degraded).

Gavrilescu et al. [18] and Rylko-Polak et al. [29] showed that reusing biomass to prepare biocomposites can be consistent with economic goals. Todor et al. [25] reached similar conclusions when considering the use of natural-fiber-based composites in textile applications. Krauklis et al. [15], in their study of the end-of-life options for high-performance composites, reached the conclusion that both profitability and a reduction in GWP would usually benefit from reuse and recycling practices. When comparing options for used composites, the environmental impact of recycling was reported to be only about 15% to 28% that of either landfilling or incineration, with respect to climate change, the use of limited resources, ecosystem quality effects, and human health [15]. Such savings are achieved

through the avoidance of processing new materials for the intended uses. For example, the amount of energy required to melt and re-form a thermoplastic-based material, whether or not it contains reinforcing fibers, can be a small fraction of the energy required to prepare fresh materials.

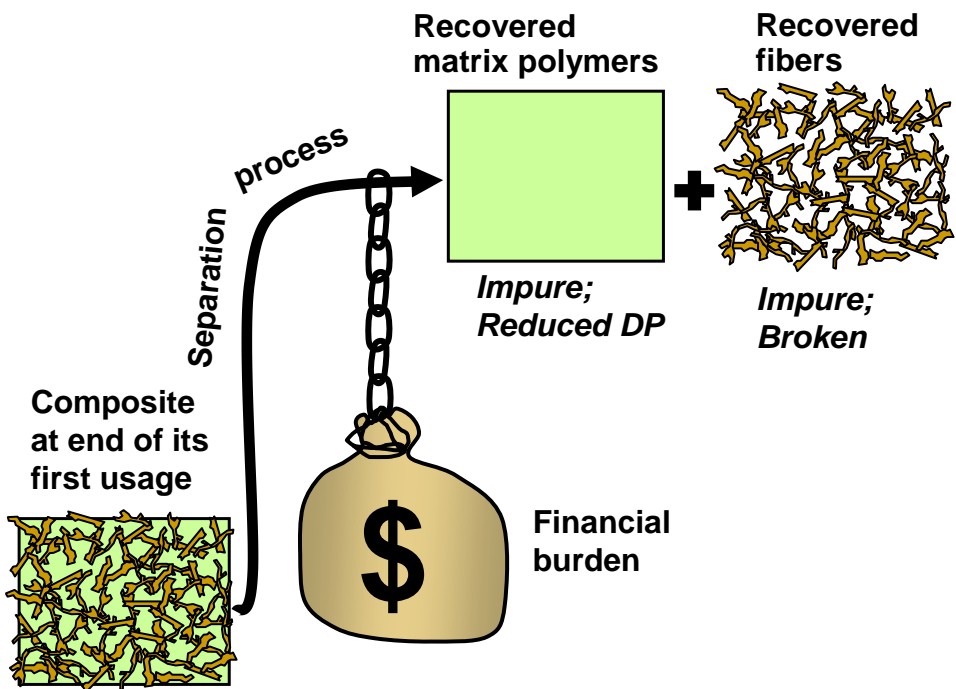

**Figure 4.** Emphasizing the financial burden that often stands in the way of implementing processes involving the separation of the components of a composite.

## 4. Does the Product Last a Long Time?

### 4.1. Composite Strength and Durability

Certain composite structures are known for achieving high strength and durability [30–32]. The importance of such strength advantages, in terms of environmental impacts, is not always appreciated. Studies have shown that long-lasting consumer items can provide a net reduction in GWP in comparison to similar items that require more frequent replacement [33]. Figure 5 shows the replotted output from a dynamic life-cycle analysis, in which different product lifetimes (before incineration) and different rotation periods for the wood utilized to make a glulam lumber product [34]. The figure shows the calculated results for the three selected cases that were reported in that article. The pink area represents the (unfavorable) net effect on GWP when the assumed product life is short (1 year), and the assumed growth period of the trees is very long (200 years). The green area represents the (most favorable, "carbon negative") net effect of the glulam production practices when the assumed product life is relatively long (150 years), and the growth period for the wood is relatively short (50 years). The orange area (partly hiding parts of the pink and green areas) represents an intermediate case. As shown, the greatest benefits in terms of GWP were achieved when the glulam product was kept in service for a long time (e.g., 150 years), and the tree rotation period was relatively short (50 years in this example). Assuming that the item is not perceived as being outdated and therefore discarded, the ratio of manufacturing-related environmental impacts becomes spread out over a longer useful lifetime.

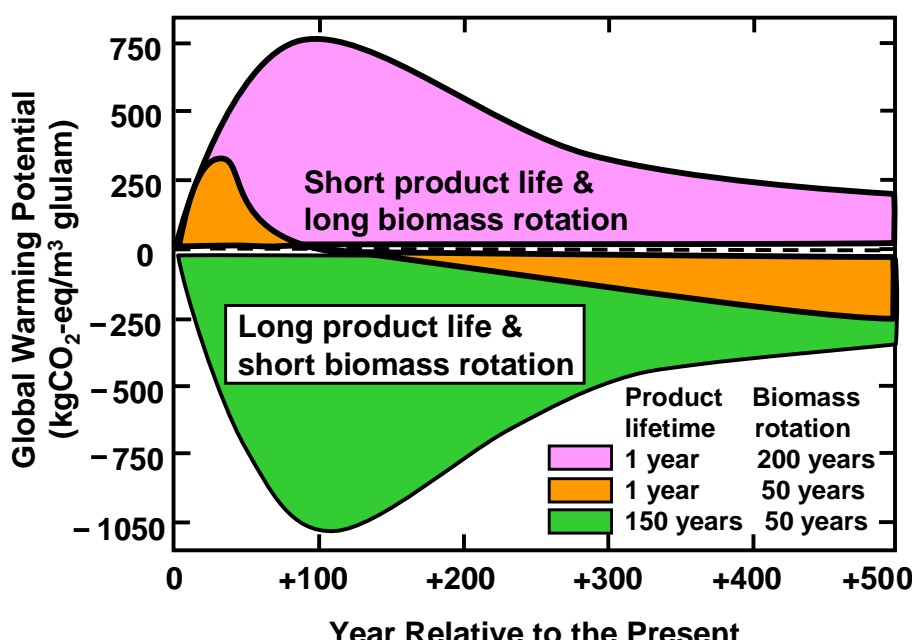

**Figure 5.** Selected output from dynamic life-cycle analysis of glue-laminated (glulam) products with different assumed product lifetimes before incineration and different rotation periods of the trees from which the wood is obtained (elected data replotted from Cardellini et al., 2018) [34].

When items are intended for long-term usage, life-cycle principles will favor materials and methods that maximize the useful lifetime. To return to the example of wind turbine blades, very high mechanical strength can be achieved by (a) using premium quality carbon fiber; (b) forming a suitably uniform or intentionally aligned mat of the fibers; (c) ensuring uniform impregnation with a curable resin, such as epoxy; and (d) curing the resin [15]. Such an approach avoids the undesired clustering of fibers in cases where researchers attempt to mix fibers, even relatively short ones, at levels of more than 2% to 5%, in polymer melts [17].

### 4.2. Fiber Surface Modification and Compatibilizing Agents

When cellulosic fibers are employed as reinforcements in the most widely used thermopolymer matrices, the fibers' polar, hydrophobic nature can inhibit or even prevent good molecular contact between the phases [3,4,19]. Nassar et al. [3] outlined four basic approaches to overcoming such issues, namely mechanical interlocking, chemical bonding, molecular inter-diffusion (when the two phases are chemically compatible), and electrostatic bonding. In general, one can focus either on the modification of fiber surfaces or adding something to the matrix material that is capable of either reacting with –OH groups at the surface of biomaterials. Either of these approaches can help the two surfaces to spread over each other, at a molecular level, and achieve good adhesion during melt forming [35]. The reported approaches to improving contact between biobased reinforcements and hydrophobic matrix materials have been highly diverse, and in typical cases, they can increase strength properties by a factor of as much as two [17].

In principle, the ability of two phases to spread over each other at a molecular level, when at least one of the phases is in liquid form, can be predicted based on solubility principles. It is well known that interfacial wetting and some inter-penetration of macromolecular segments at the interfaces will be favored when the two facing surfaces are more similar to each other [3,4,35]. The key parameters that have been used to judge such similarity include the cohesive energy density (Hildebrand parameter), the polar character, and the hydrogen bonding ability of the surface layers of each facing surface. Widely utilized plastics, such as polyolefins and polyesters, tend to be non-polar, with relatively low Hildebrand parameters

compared with typical lignocellulose-based particles. As a consequence, composites made with such plastics reinforced with unmodified cellulose-based reinforcements often show unfavorable mechanical strength values [17]. Electron micrographs of such composites often show a lack of contact between the contrasting phases [3]. Void areas between the matrix and reinforcing fibers also can develop in the course of aging of a composite [31].

A wide range of approaches have been demonstrated for the surface treatment of plant-based fibers used as reinforcements in composites [36]. Of these, one of the most commonly employed, for improvements in manufacturing outcomes, has been with alkaline aqueous solutions [37,38]. By removing monomeric and loosely held material from fiber surfaces, it is often possible to achieve better contact between the polymeric parts of the reinforcing fibers and the matrix material. In addition, the alkaline treatments may remove some of the hydrophilic hemicellulose from the material. Researchers also have employed various chemical approaches to graft hydrophobic alkyl chains or other hydrophobic groups onto plant fiber surfaces to make them more compatible with hydrophobic matrix polymers [38–40]. As described in the cited articles, promising chemical routes include esterifications, reactions of alkyltrialkoxysilanes, and certain plasma treatments. Though such treatments generally improve the bonding between the phases in a composite, it is worth keeping in mind that plant-based fibers, even those with a reputation for strength, such as jute and hemp, are not nearly as strong as carbon fibers [15]. When excellent wetting and adhesion between the matrix polymer and reinforcing fibers are achieved, the strength of the fibers may become the strength-limiting factor.

### 4.3. Composites Intended for Single Use

Single-use products, such as many kinds of food packages or serving items, are not intended to be durable. For such items, the only viable approach in the direction of sustainability is to keep any environmental impacts as low as possible. A paper plate, for instance, can be regarded as competing for sustainability status against rivals that include ceramic and glass plates, each of which can be used multiple times. At the front end, relatively low environmental impact can be achieved with the use of plant materials, which typically have a lower carbon footprint than most alternatives [20]. At the terminal end, used paper plates are often suitable for composting, whereby the materials can be returned to the soil in a beneficial form [41]. As noted by Maiti et al. [32], a balance between durability and biodegradability needs to be achieved, depending on the end-use application of the composite. Most sanitary paper and paper that has been in contact with food are not currently recycled, which is probably mainly due to concerns about contaminants, as well as moist conditions that can lead to rot. Unless that situation changes in the future, there will continue to be a motivation to keep environmental impacts as low as possible during manufacture and to route the waste materials toward the production of soil amendments (composting) or other beneficial uses involving circular economy principles (see next section).

## 5. Are Circular Economy Goals Achieved?

### 5.1. Making Sure That Every Byproduct or Discarded Item Gets Reused

The term "circular economy" envisions a socio-economic system in which every byproduct of manufacture or residue at the end of current usage is ultimately recycled to take advantage of any residual value [30]. One way to picture this is to assume that when no current value remains, perhaps due to structural and chemical breakdown during usage, the material ultimately biodegrades and is returned to the natural cycle of life, e.g., as part of the soil [26,42]. The principle is illustrated schematically in Figure 6.

Ayre [30] noted that the plant materials employed in making various composites are compatible with that kind of circular economy since they are fully biodegradable. However, it was also noted that a lot of work remains to be done to advance the state of the art of forming fully biobased matrix materials that will be able to replace petroleum-based polymer materials, as fully biobased composites become increasingly common in the future.

Ayre [30] also proposed an approach of designing for "controlled decomposition" when the item is landfilled or composted at the end of its current usage.

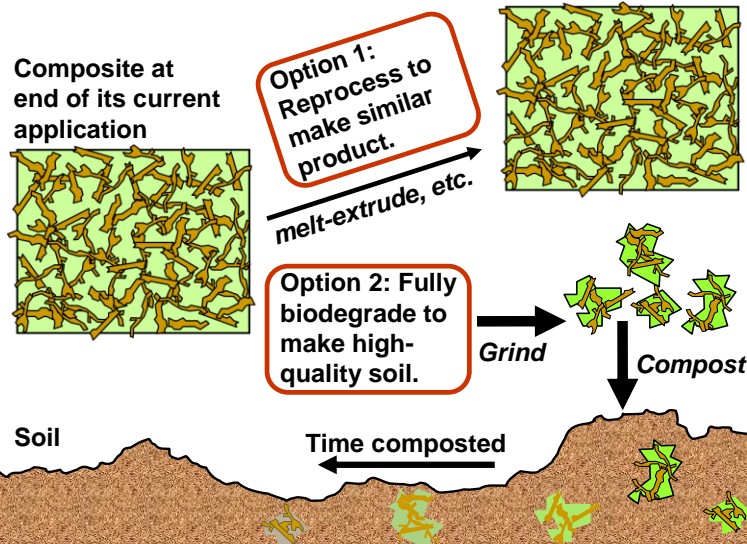

**Figure 6.** Two circular economy options when a biodegradable composite comes to the end of its usefulness in its current form.

### 5.2. Can the Material Be Recycled without Major Loss of Value?

Poly(lactic acid) (PLA) and other biobased plastic materials are known to suffer from the breakdown of their polymeric form each time that they are melted for compounding and extrusion into a new generation of product [35,43]. To some extent, these tendencies can be compensated by chain-extension treatments, which are intended to bolster the polymer molecular mass during reprocessing [44,45]. Further improvements can be achieved with the use of additives in the original composite that minimize damage to polymers during reprocessing [46].

Another challenge lies in the degradation of plant-based fibers during the thermal reprocessing of thermoplastic composites. Such reprocessing appears well suited for thermopolymer composites reinforced by carbon or glass fibers, which are not adversely affected at temperatures high enough to melt commonly used plastics [47]. Fiber breakage and thermal degradation can occur when the corresponding composites contain cellulosic fibers [48–50]. At present, there does not appear to be a way to reverse the progressive degradation and shortening of cellulosic fibers that can be expected to accompany melt-processing by means of heating and extrusion.

### 5.3. Is the Whole Composite Truly Biodegradable?

In view of the progressive degradation of both bioplastics and cellulose-based reinforcing fibers during melt reprocessing, as just discussed, it becomes even more important to make sure that the composite material is capable of being thoroughly biodegraded at the end of its current usefulness [51]. A key concern is that the most widely reported biopolymer, PLA, is not known to be biodegradable under typical conditions in soils or in the sea [35,52]. Rather, the mechanism appears to depend on abiotic processes that require temperatures of about 55 °C or higher. Such conditions can be readily achieved in industrial composting systems but not in landfills or when the items are cast out as litter.

Another approach is to employ wood-derived lignin and other biopolymers such as tannins or starch products as the matrix phased in fully biobased composites [53,54]. Oliaei et al. [54] showed that the modulus of rupture and the modulus of elasticity achievable through the hot pressing of lignin-containing natural fibers can greatly exceed that of ordinary paper and ordinary fiberboard products. A likely downside of such strategies

would be lower strength performance in comparison to the highly optimized petroleum-derived plastics or resins formed under similar conditions of pressing. However, that situation may change to some extent due to continuing developmental progress with nature-based adhesives [26,54]. In each case, there may be a balance that needs to be achieved between strength performance and biodegradability.

## 6. Are Cascade Principles Followed?

### 6.1. Examples of the Cascade Principle

The word "cascade" can be interpreted in different ways. For instance, Gravrilescu et al. [18] stated that "Cascade utilisation is designed to optimally use all components of the materials, including multiple usages of chemical resources, before they are used to generate power at the end of their life cycles". Further benefits can accrue if, after each successive stage of usage, the highest possible value is obtained [55]. For example, Lubke et al. [56] considered successive stages of reuse involving particleboard, medium-density fiberboards, and low-grade linerboard paper. Thus, it has been proposed to think in terms of a waste hierarchy, in which a succession of reuse pathways are employed, though the final disposal may still involve combustion and conversion to energy [57]. Krauklis et al. [15] considered a system in which used material would be sorted into two categories, the higher value of which would be directly used for similar purposes as the first generation of composite, and the second category for lower uses, such as fillers for composites or recycled fibers for various non-demanding applications. Geldermann et al. [58] emphasized the importance of multiple utilizations of material before its ultimate conversion into energy by incineration. Liu et al. [59] developed a concept in which components of composites could be pretreated in such a way as to be favorable for a cascading sequence of subsequent applications.

Figure 7 illustrates a hypothetical system in which lumber pieces and large beams, at the end of their primary usage, may be considered for particleboard or other engineered wood, possibly after the material has been used more than once in its solid-wood format. Likewise, the particleboard, in principle, can be recycled to another generation of particleboard before it might be either incinerated or used in a low-grade fiberboard or paper application. Nanocellulose could, in principle, be made from recovered paper, especially if it is intended for low-cost applications, such as viscosity control in cement or oilfield drilling operations. In the figure, the large gray downward arrows indicate the cascade from higher-value uses to lower-value uses. The circular arrows indicate reuse or recycling of the material for a similar cycle of use. The orange-shaded box towards the right represents a referred final step according to the cascade principle.

Some types of discarded items, which potentially could be incorporated into composites, are presently underutilized. The textile industry generates large amounts of fibrous waste not only during manufacture but also when clothes are discarded. In the interest of achieving circular economy goals, Echeverria et al. [60] proposed using such textile-derived fibers in composites. Sormunen and Karki [61] proposed the recycling of construction demolition materials as potential fillers for composites. There are also opportunities to incorporate cellulosic materials, either freshly harvested or recycled from other uses, into additive manufacturing (i.e., 3D printing) applications [62].

Mair and Stern [33] noted that although the terms "cascade principle" and "circular economy" share many attributes in common, they are not synonymous. The essentials of a circular economy require only that each item is either completely biodegraded or used as a material for another product at the end of its current use [63]. The cascade principle emphasizes multiple uses and specifically considers incineration as an endpoint.

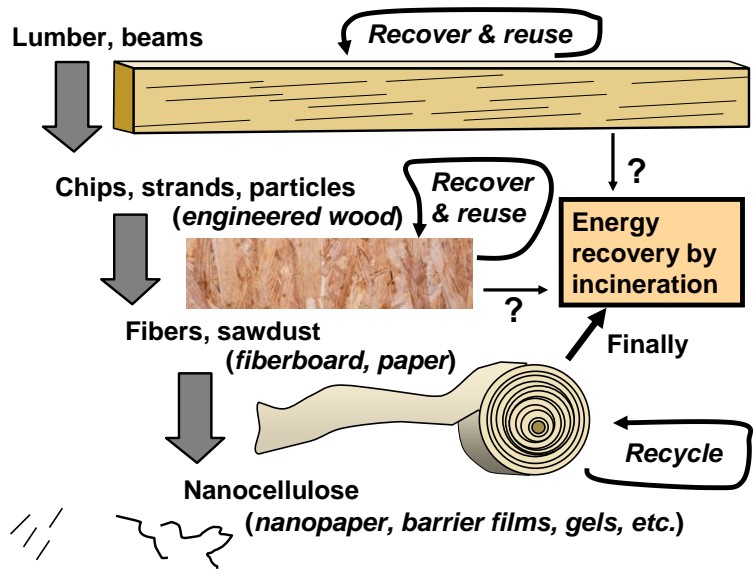

**Figure 7.** Hypothetical illustration of a cascade system, whereby larger, more valuable wood items are preferentially reused for the same purpose at this end of each cycle of use, and thereafter recycled to the next less demanding application, and ultimately incinerated as a means to generate energy.

### 6.2. Can Large Pieces of the Product Be Separated and Reused?

Relatively large, continuous parts of a manufactured item often have higher inherent value than smaller pieces or chemical components. Thus, large beams have a higher value per mass than smaller lumber pieces, strands, wood chips, and sawdust. Taskhiri et al. [64] found that, when applied to wood utilization, a cascade system of material flows achieved similar costs to the utilization of only fresh wood, but $CO_2$ emissions were greatly decreased. Hubbe and Grigsby [17] reviewed the literature related to cellulosic particles of dramatically different sizes as potential reinforcements for various polymer composites. When focusing just on the strength properties and elastic moduli of the resulting composites, no significant advantage was found for any category of size, ranging from cellulose nanocrystals to wood strands. At the same time, the costs associated with production and optional surface modification would be greatly reduced when using larger pieces.

As a high-end example of the same principle, the large sections of used wind turbine blades will have a much higher value in their intact form, compared to separating them into their components [65,66]. Thus, when mechanical properties are high, it can be advantageous to dissemble and refabricate the structure, using relatively large pieces. A similar conclusion was reached by Utekar et al. [67] when considering the most advantageous end-of-life uses of thermoset polymer composites. They concluded that the best strategy is retaining the large intact pieces, thus retaining the structural value of the material and delaying its eventual downgrade to the status of fuel for energy generation.

### 6.3. Opportunities Related to Sorting Technologies

A need for advanced sorting technologies and related sensors and infrastructure can be expected to pose large challenges as society moves in the direction of implementing higher-value reprocessing of discarded items. Fortunately, with the advancement of computers and sensors, it becomes increasingly feasible to sort out and send each discarded item to the site of its most suitable reuse or recycling operation [68–70]. Another approach, taking advantage of RFID systems and Internet connectivity, could be used to direct durable items back to their original manufacturer or a specialized reprocessing factory when the owner no longer regards them as worth keeping [71,72]. Suppose, for instance, that a discarded wind turbine blade arrives at a waste recovery site. Such items could be automatically identified and sent to specialized sites for disassembly and manufacture of other durable items, through cutting and reassembly. Other manufacturers could specialize in the reuse

of recovered lumber pieces, which already have high value in venues where a rustic appearance is desired [73].

### 7. Is Reuse Highly Likely, Not Just Theoretically Possible?

#### 7.1. Already Implemented Technologies

When deciding where to allocate limited research dollars, it makes sense to focus on recovery technologies that have a high probability of being implemented. The best initial place to look for such technologies is within existing companies that manufacture such products as particleboard [74]. The use of scraps from other industries represents the main strategy in such product lines. More reliance on recovered used materials, though it would involve complications, seems likely to expand with the passage of time. Specific challenges to be met include what to do about nails, as well as problems due to paint, antifungal treatments, and other contaminants that may affect various potential applications. Figure 8 indicates three hypothetical uses of cellulose-based material after it has reached the end of its useful service in its present form. Such uses include coarse chipping, leading to its potential usage in particleboard; fiber production (e.g., chemical or mechanical pulping), leading to potential usage in a low-grade paperboard product; or breakage into coarse pieces that might be suitable for road fill. Note that this figure emphasizes approaches that do not require the isolation of the matrix material from the composite but rather process the entire material to achieve another valuable use.

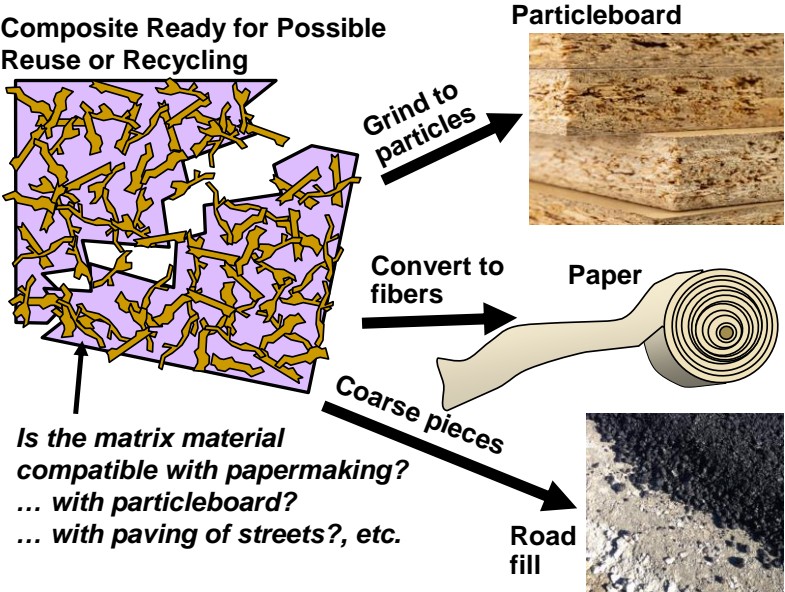

**Figure 8.** Some possible end-of-life options for sustainable composites that may not require separate recovery of the matrix.

#### 7.2. Papermaking

Another highly promising route to reprocessing certain composite materials is papermaking, especially when considering low-quality paperboard or molded products, such as egg cartons. Such packaging products can be favored when legislation blocks the usage of expanded Styrofoam in such applications [75]. In principle, papermaking fibers can be used multiple times just by redispersing the material in water, forming a new generation of wet paper, drying it under tension, and then applying such processes as surface starch, coatings, and smoothing (calendering) via the passage between smooth cylinders [76]. Certain contaminants such as plastic laminate material can be readily separated from the fibers during conventional recycling of recovered liquid containerboard, such as used milk cartons [77].

### 7.3. Challenges in the Recovery of Matrix Polymers

The proportion of synthetic plastics that become recycled is relatively low. For instance, Datta and Kopczynska [78] reported that only about 26% of European post-consumer plastics were being recycled back to plastic items, whereas about 35% were used directly for energy generation, and about 38% were being landfilled. The reasons for the relatively low recycling levels include the relatively low costs of petroleum in recent years, in addition to the maturity of technologies and economies of scale associated with making plastics from petroleum resources [79]. By contrast, efforts to utilize recycled plastics face challenges due to contaminants, variability in the quality of materials, and the progressive breakdown of polymers in the course of multiple reprocessing cycles [35,43]. The situation can be expected to become even more daunting when such used synthetic plastic materials are in the presence of reinforcing fibers, whether they be biobased or otherwise. Though the recovery of such plastic material from composites has been shown to be possible [80,81], one would not expect that such processing would become economically attractive within a socio-economic world where large amounts of single-component plastics still fail to be recovered from waste streams.

### 7.4. Biorefinery Systems

Another approach, still related to the "keep it simple" principle, is based on the conversion of the material back to essentially monomeric form. The word "biorefinery" has been used when the goal is to obtain such compounds from biobased materials, which often are a main component of sustainable composites [82,83]. Though each individual compound obtained in a biorefinery system might be regarded as being simple, the mixture, as well as the steps needed to isolate the compounds in relatively pure form, will be complex. In principle, the recovered compounds from a biorefinery could replace compounds newly obtained through the conventional refining of fossil resources [84]. As an example, Zia et al. [80] discussed the glycolysis of polyurethanes to recover the monomers from used polymeric material. However, those authors concluded that energy recovery by incinerating the plastic material, or its composites, was more practical. Another way to recover biopolymers, as noted by Kostag and El Seoud [85], involves their dissolution in suitable solvents, including ionic liquids. However, Shuaib and Matifenga [10] concluded that solvent usage can have a dominant negative environmental impact. In general, the chemical-based recycling of composites can be expected to be more dependent on energy in comparison to alternative approaches [81].

Presently, it appears that the largest barriers to the development of biorefinery operations include a relatively low prevailing cost of crude petroleum, greater variability, and higher oxygen content of typical biomaterials, compared with petroleum resources, as well as the highly optimized and built-up status of current manufacturing systems based on petroleum resources. However, there appear to be opportunities to integrate biorefining into some existing facilities presently used for refining petroleum products [84,86].

### 7.5. A Potential Competitor—Self-Reinforcing Polymers

When envisioning the likely advantages of well-engineered sustainable composite systems, there is a danger that one might ignore potential competing systems that have the potential to render such efforts obsolete. To give one example, progress has been achieved in the fabrication of so-called single-component polymer composites or self-reinforced polymer composites [87–90]. By taking advantage of contrasting processing conditions, the same polymer chemistry can be used to achieve effects that previously could only be achieved when using two or more different materials in combination. With fine-tuning aspects of preparation, including such factors as orientation and annealing, significant gains in mechanical properties can be achieved [87,90]. As noted by Matabola et al. [89], single-polymer composites are much easier to recycle than ordinary composites that are reinforced with solid particles. In effect, they can achieve a key benefit

of ordinary composites (improved properties) while avoiding a key detriment (lack of expected demand at plastic reprocessing facilities).

## 8. Conclusions

This review article involves a series of critical questions to be asked by scholars who are either proposing or carrying out research related to the topic of sustainable composites. The consideration of such questions can help ensure that research efforts have a good chance of transforming the state of technology, leading to the needed reductions in GWP and other environmental impacts of modern society. To begin, it is important to focus on the expected carbon footprint of each proposed technology. In other words, what is the expected effect of implementing such technology on the net emission of greenhouse gases? There is some potential to reduce GWP through the use of sustainable composites, but the results are likely to depend on such issues as the compatibility between the phases, as well as the amounts of energy used during the growing, transporting, and processing of the biomass. Researchers also need to ask themselves about other potential life-cycle impacts, such as eutrophication, ozone layer depletion, acidification, and various aspects of toxicity. It is also important that the proposed technological approaches have favorable cost structures, keeping in mind that economies of scale may help to bring down costs as processed volumes increase.

LCA studies have shown that the most favorable results, in terms of environmental impacts, can be achieved when manufacturing items that last for a long time. In that way, any adverse environmental impacts can be spread out over many years of service. The biobased content of such products can have a favorable net impact and GWP, since that portion of the carbon content comes from photosynthesis. The ability of cellulose-containing composites to achieve sufficient strength and durability to last many years can be enhanced through certain chemical modifications, often involving the modification of the cellulosic surfaces.

The principles of circularity and cascading can be applied as ways to integrate the use of certain composite materials into a sustainable future. A classic concept of circularity may include the full biodegradation of the material at the end of its useful life. For example, the composite might be converted to a useful soil amendment via a composting process or equivalent biodegradation. But such a concept takes a long time, and not every composite structure has a reasonable chance of becoming fully biodegradable. Thus, it makes sense to emphasize options involving the reuse and recycling of materials at the point where the initial composite material is no longer needed for its first usage. The word cascading, in the most favorable circumstances, can mean that the material is employed for its next most valuable potential usage. At the top of the chain, the item could be disassembled, allowing for the reuse of the main structural parts. At intermediate levels, the composite material could be reduced in size and employed in applications such as particleboard. And finally, when there are no other potential uses, the energy content can be captured via an efficient process of combustion. In each case, it is important to keep in mind a final question about the likelihood that a proposed technology will actually become implemented. It is only when the proposed technological approaches become widely implemented that it is reasonable to expect a net contribution to solving some pressing environmental problems that are facing our society.

Based on the articles cited in this review of the literature, it can be stated that much progress has been achieved in the development of sustainable composites. Extensive research still needs to be performed in the coming years, as people attempt to deal with serious issues related to the environmental impacts of technology. When selecting approaches for future research, emphasis needs to be placed on potential solutions that simultaneously address carbon footprint (global warming issues) and other life-cycle issues, achieve suitable levels of durability, allow for the full reuse or recycling of components, retain the material's high value in the course of multiple uses, and have a high likelihood of implementation at a large scale.

**Funding:** The research work of Martin A. Hubbe is supported by an endowment from the Buckman Foundation.

**Institutional Review Board Statement:** Not applicable.

**Informed Consent Statement:** Not applicable.

**Conflicts of Interest:** The author declares no conflict of interest.

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
