# Peer review of "Sustainable Composites: A Review with Critical Questions to Guide Future Initiatives"

_sustainability, doi:10.3390/su151411088_

Round 1

Reviewer 1 Report

In this work, the author reviews some published work on sustainable composites and summarizes the conditions under which they can be made sustainable, as well as some of the problems that may arise, and put forward some suggestions on the direction and focus of sustainable composites in the future. The whole manuscript is detailed and logical, but still need some minor revised before it may be considered further for publication in Sustainability. Some comments and concerns and questions are given below that need to be addressed by the authors:

(1)   Each paragraph of the introduction is too independent to reflect the meaning of the work, so it needs to be revised carefully to highlight the main points of this article.

(2)   The explanation of Figure 5 in the manuscript is unclear, the meaning of each part need to be further expressed.

(3)   There are still some errors and deficiencies in format and expression in the manuscript. For example, in Section 2.2, the author does not explain which two ways could reduce the global GWP, and in Line 383, what is the “highlight method in Figure 8”.

(4)   Many of the statements in this manuscript need more evidence to prove. Such as in Line 84 to 91, “that addition levels above a certain threshold, often in the range of 2 to 5%, leads to excessive clustering of the particles… …This results in decreased strength beyond that threshold.”, and in Line 210 to 219, how the researchers “graft hydrophobic alkyl chains or other hydrophobic groups onto plant fiber surfaces to make them more compatible with hydrophobic matrix polymers.”.

(5)   In Line 197, Part 4.2, as the other method to improve the compatibility of cellulose and matrix materials, how to solubilization of matrix materials need to be further introduced.

(6)   It is hoped to make a comprehensive comparison of the comprehensive properties of different composite materials mentioned in the paper, such as carbon fiber reinforced composite materials, glass fiber reinforced composite materials, cellulose reinforced composite materials, such as mechanical properties, GWP, average service life, etc., list the corresponding data, and have a more intuitive comparison and explanation.

(7)   The Conclusion part failed to summarize and generalized of the full manuscript, need to be rewritten.

The English used is correct and readable.

Author Response

Please see the attachment.  I want to make sure that the colored highlighting and italics can be visible.

Reviewer 2 Report

The paper discusses the definition of sustainability in composites. It is very essential for current research on sustainable composites, which is an important research direction.

I supports the publication, but authors need to have some discussion on the following type of polymer composites, often recognized as self-reinforced polymer composites, single polymer composites, polymer-fiber-reinforced polymers, or polymer fiber – polymer matrix composites.

This type of polymer composites as well as natural fiber composites (as mentioned in the original manuscript by the authors) are considered as substitution of carbon-/glass-fiber reinforced polymers in some cases for the purpose of recyclability, sustainability, and others. The following are some literature review articles that authors can refer to and further strengthen the work:

1.       Kmetty A, Bárány T, Karger-Kocsis J. Self-reinforced polymeric materials: A review. Progress in Polymer Science 2010;35:1288-310. https://doi.org/10.1016/j.progpolymsci.2010.07.002

2.       Qiao Y, Pallaka MR, Fring LD, Simmons KL. A Review of the Fabrication Methods and Mechanical Behavior of Thermoplastic Polymer Fiber - Thermoplastic Polymer Matrix Composites. Polymer Composites 2023;44(2):694-73. https://doi.org/10.1002/pc.27139

3.       Matabola KP, De Vries AR, Moolman FS, Luyt AS. Single polymer composites: a review. J Mater Sci 2009;44:6213-22. https://doi.org/10.1007/s10853-009-3792-1

4.       Alcock B, Peijs T. Technology and development of self-reinforced polymer composites. In: Polymer composites-polyolefin fractionation-polymeric peptidomimetics-collagens. Advances in Polymer Science. Heidelberg, Springer, Berlin, 1–76, 2011

Author Response

See the attached file.  I want to make sure that the colored highlighting and the italics can be seen.

Reviewer 3 Report

Martin A. Hubbe in the manuscript entitled "Sustainable Composites: A Review with Critical Questions to Guide Future Initiatives", reviewed the required conditions to obtain a structure that can be called sustainable. The review discussed some questions related to the carbon footprint, other life-cycle impacts, durability, recyclability without major loss of value, reusability of major parts, and the practical likelihood of various end-of-life options. However, some of the deficiencies listed below were observed:

  1. The manuscript contains some typos/spelling mistakes. They should be corrected.
  2. The novelty of this review is absent. The article contains general information. Why should the readers read this review??
  3. Section “2.2. Reduction of Product Weight” is weak. The author should add more information with related references in order to strengthen this section.
  4. The author should add more references and information with a better comparison between the methods used from the point of view of LCA. Section "2.3. Bio-based Content" contains general information. The author should add specific information.
  5. In section “3.2. Economic Impact”, the author wrote “Krauklis et al. [15], in their study of the end-of-life options for high-performance composites, reached the conclusion that both profitability and reduction of GWP would usually benefit from reuse and recycling practices.” Why? The author should summarize the research results and conclusions.
  6. The aim of section “4. Does the Product Last a Long Time?” is absent. All presented information is known. The author should explain his point of view.
  7. Why are the “Cascade Principles” important? How are they related to Future Initiatives?

The article contains some typos/spelling mistakes. Minor editing of English language required.

Author Response

Please see the attachment.  I want to make sure that the colored highlighting and italics can be seen.

Round 2

Reviewer 2 Report

I am satisfied with the revision. It is ready for publication.

Reviewer 3 Report

The author made sufficient improvements to the manuscript.